# Comparative Binding Ability of Human Monoclonal Antibodies against Omicron Variants of SARS-CoV-2: An *In Silico* Investigation

**DOI:** 10.3390/antib12010017

**Published:** 2023-02-23

**Authors:** Nabarun Chandra Das, Pritha Chakraborty, Jagadeesh Bayry, Suprabhat Mukherjee

**Affiliations:** 1Integrative Biochemistry & Immunology Laboratory, Department of Animal Science, Kazi Nazrul University, Asansol 713 340, India; 2Department of Biological Sciences & Engineering, Indian Institute of Technology Palakkad, Palakkad 678 623, India

**Keywords:** Omicron, SARS-CoV-2, Variants, monoclonal antibody, chimeric mAb, molecular dynamics simulation, immunotherapy

## Abstract

Mutation(s) in the spike protein is the major characteristic trait of newly emerged SARS-CoV-2 variants such as Alpha (B.1.1.7), Beta (B.1.351), Gamma (P.1), Delta (B.1.617.2), and Delta-plus. Omicron (B.1.1.529) is the latest addition and it has been characterized by high transmissibility and the ability to escape host immunity. Recently developed vaccines and repurposed drugs exert limited action on Omicron strains and hence new therapeutics are immediately needed. Herein, we have explored the efficiency of twelve therapeutic monoclonal antibodies (mAbs) targeting the RBD region of the spike glycoprotein against all the Omicron variants bearing a mutation in spike protein through molecular docking and molecular dynamics simulation. Our *in silico* evidence reveals that adintivimab, beludivimab, and regadanivimab are the most potent mAbs to form strong biophysical interactions and neutralize most of the Omicron variants. Considering the efficacy of mAbs, we incorporated CDRH3 of beludavimab within the framework of adintrevimab, which displayed a more intense binding affinity towards all of the Omicron variants viz. BA.1, BA.2, BA.2.12.1, BA.4, and BA.5. Furthermore, the cDNA of chimeric mAb was cloned *in silico* within pET30ax for recombinant production. In conclusion, the present study represents the candidature of human mAbs (beludavimab and adintrevimab) and the therapeutic potential of designed chimeric mAb for treating Omicron-infected patients.

## 1. Introduction

Since the first report of the Severe Acute Respiratory Syndrome Coronavirus 2 (SARS-CoV-2) outbreak in December 2019, the virus continues to threaten mankind with high-grade morbidity and mortality [1,2,3,4]. Owing to its higher rate of infectivity, rapid transmission and severity in causing death, the World Health Organization officially declared COVID-19 as a pandemic on 11th March, 2020 [5]. Clinical trials of several repurposed (hydroxychloroquine, doxycycline, flavipiravir, and ivermectin) and/or novel drug candidates (2-deoxy glucose) as well as the administration of several newly developed vaccines (viral vector, mRNA, inactivated whole virion, attenuated, and subunit vaccine) are found effective in reducing the pathogenic attributes of SARS-CoV-2. However, variants of SARS-CoV-2 such as Alpha, Delta and Delta-plus also emerged within a short time frame. These strains possess mutations across the whole genome, including in the spike protein, and are characterized by a higher rate of infectivity, pathogenesis, and death-inducing ability, as well as the ability to escape protective immunity elicited by previous infection and vaccines [3,4,5,6]. The Omicron variant (B.1.1.529) is most critical as it contains above 30 different viruses bearing mutations (substitutions, deletions, and insertions) in the spike glycoprotein [6,7,8]. Such mutations in the spike protein cause greater transmissibility compared to the other strains of SARS-CoV-2 [6,7,8]. Within four months of its emergence, Omicron was found to be the dominant SARS-CoV-2 strain across 188 countries [9]. Most surprisingly, this variant has the ability to escape the protective host immunity induced by either natural preinfection or by vaccination [10,11,12]. 

SARS-CoV-2 possesses a nonsegmented positive-sense RNA genome of 30kb along with the viral spike glycoprotein, membrane, and envelope surface viral proteins [13]. The RNA genome encodes four structural proteins, namely membrane glycoprotein (M), nucleocapsid (N), envelope (E), and spike protein (S), along with nonstructural proteins (NSPs) such as main protease (Mpro), RNA-dependent RNA polymerase (RdRp), and nine accessory proteins (Orf3a, Orf3b, Orf6, Orf7a, Orf7b, Orf8, Orf9b, Orf9c, and Orf10) [14]. Viral entry into the host’s target cells and active infection is totally dependent on the binding of spike glycoprotein with angiotensin-converting enzyme 2 (ACE2) [15]. Priming of the spike protein by serine protease TMPRSS2 and endosomal protease Cathepsin L splits the spike proteins into S1 and S2 subunits. The S1 subunit is composed of one N-terminal domain (NTD), one receptor binding domain (RBD), and two C-terminal domains (CTD) [16]. Successful fusion of the S2 subunit with the cellular membrane leads to viral entry within the host cytoplasm [16]. The spike glycoprotein also serves as a ligand for the human toll-like receptor four (TLR4) that in turn triggers classical NF-κβ activation to induce the expression of proinflammatory cytokines and disruption of inflammatory homeostasis of the host leading to hyperinflammatory consequences, failure of the vital organs, and death [17,18,19].

Hitherto, a total of 30 mutations have been documented in the spike glycoprotein of Omicron wherein 15 are located within the RBD [20,21]. Amongst the mutations in RBD, four are common for all the Omicron clades viz. BA.1, BA.2, BA.2.12.1, BA.4, and BA.5. Each mutation has been deciphered for the inclusion of important attributes for survival and immune escape [22,23,24,25,26]. For example, mutation at the E484 and K417 positions resulting substitution of A with N has been found to increase resistance to antibody-mediated virus neutralization [22,23,24,25,26]. On the other side, N501Y substitution enhances transmissibility while T478K mutation strikingly enhances the sensitivity towards the therapeutic monoclonal antibodies (mAbs) [22,23,24,25,26]. Apart from these, nine more mutations (Y505H, N501Y, Q498R, G496S, Q493R, E484A, S477A, G446S, and K417N) have been documented at the ACE2-binding region [27,28]. These mutations alter the binding affinity of RBD with ACE2 and that in turn alters the virulence of the Omicron variants [29]. Intriguingly, the available treatment approaches exhibit limited efficacy against the aforesaid strains that have a rapid potential to establish infection and transmission thereafter.

In this context, therapeutic human mAbs are currently being employed as alternative therapeutics to target the RBD region of spike glycoprotein. In our previous study, bamlanivimab, regdanvimab, tixagevimab, cilgavimab, etesevimab, casirivimab, imdevimab, and sotrovimab have been investigated for their efficacy to bind the spike glycoprotein of newly emerging Alpha, Delta, and Delta-plus strains of SARS-CoV-2 [30]. However, most of the therapeutic mAbs have failed to prove their efficacy against the newly emerged Omicron strain while some discrete evidence has shown a limited activity of mAb cocktail as well as combination therapies. In this report, we have investigated a total of 12 human therapeutic mAbs (having potential therapeutic activities against the Alpha, Delta, and Kappa strains of SARS-CoV-2) for their binding affinity against the spike protein mutants of the Omicron clade. In addition, we have also presented a novel chimeric mAb by combining the antigen-binding paratopes of the active mAbs for treating Omicron strains.

## 2. Materials and Methods

### 2.1. Data Mining

Amino acid sequences corresponding to the spike glycoproteins from the various Omicron variants (B.1.1529) of SARS-CoV-2 were retrieved from the GISAID database (https://www.gisaid.org/ accessed on 6 June 2022). The 3D structure of each mutated spike glycoprotein was generated through a homology model using the amino acid sequence of native spike glycoprotein (Accession ID: QHD43416.1) of SARS-CoV-2 available in the NCBI database (https://www.ncbi.nlm.nih.gov/, accessed on 6 June 2022). On the other hand, the complete amino acid sequences of therapeutic mAbs were retrieved from the CoV-AbDab database (http://opig.stats.ox.ac.uk/webapps/covabdab/, accessed on 7 June 2022).

### 2.2. Homology Modelling and Validation

Homology modelling was executed for the spike proteins of the Omicron variants, namely BA.1, BA.2, BA.2.12.1, BA.4, and BA.5 bearing single mutant amino acid. Later, a unique spike protein structure named multiverse was modelled by utilizing all the mutant amino acids responsible for Omicron covariants. All the experiments to achieve the models of the mutant form were accomplished by utilizing the fully automated template-based SwissModel web tool (https://swissmodel.expasy.org/interactive, accessed on 8 June 2022) [31,32]. Next, we explored ABodyBuilder to model the Fv regions of the mAbs. ABodyBuilder (http://opig.stats.ox.ac.uk/webapps/newsabdab/sabpred/abodybuilder/, accessed on 12 June 2022) is the most advanced therapeutic antibody modelling application as it provides optimum model accuracy and it is capable of building models for complementarity-determining region (CDR) loops and nanobodies, such as the Fv regions of the antibodies [33,34]. EpiPred (http://opig.stats.ox.ac.uk/webapps/newsabdab/sabpred/epipred/, accessed on 13 June 2022) and Antibody i-Patch (http://opig.stats.ox.ac.uk/webapps/newsabdab/sabpred/antibodyipatch, accessed on 14 June 2022) webtools were utilized to determine the epitopes and paratopes by exploiting the homology models of mAbs and Omicron variants [35]. Later, to validate the stereochemical nature of 3D structures of the spike protein mutants and mAbs, the SAVES (https://saves.mbi.ucla.edu/, accessed on 20 June 2022) server was run for different parameters viz., ERRAT, VERIFY 3D, PROVE AND PROCHECK [36,37,38,39].

### 2.3. Molecular Docking and Determination of Biophysical Interactions

We exploited molecular docking for determining the protein–protein interaction, as it is a widely accepted and validated technique for examining the interaction between the spike protein mutants from the Omicron variants and the therapeutic mAbs [40,41]. HADDOCK2.4 webserver was explored to conduct molecular docking experiments for studying Omicron spike protein–mAbs interactions. HADDOCK stands for high ambiguity driven protein–protein DOCKing and enables a flexible nature of docking [42]. Ambiguous interaction restraints (AIRs) drive the docking methodology following biochemical and biophysical interaction data, resulting from NMR titration experiments. This server utilizes the PDB files of mAbs and the mutant spikes of Omicron variants and provides different docked structures that comprise the spike protein-mAb complex as output [43]. The best docked pose was determined by lowest intermolecular energy estimated by the docking procedure. All the docked files were analysed for different no-covalent interactions, such as hydrogen bond, electrostatic interaction, and hydrophobic interaction. Discovery Studio Client 2020. PyMOL was used to visualize the docked complex in space-filling mode.

### 2.4. Determination of Binding Affinity

Stable protein complexes or docked structures comprising mutant spike proteins of Omicron variants and the human mAbs were examined for biophysical stability through the PRODIGY web tool (http://milou.science.uu.nl/services/PRODIGY, accessed on 24 August 2022). Binding affinity (ΔG) and the dissociation constant (Kd) were to determined accordingly [44,45].

### 2.5. Analysis of Conformational Stability, Molecular Motion, and Dynamics

We validated the postulations drawn from molecular docking using molecular dynamic simulation which today is considered the most scientifically validated tool for examining computational predictions [46,47]. Conformational stability and topology of the complexes of mAbs with the mutant spike proteins of Omicron variants were computed by studying the dynamics of the proteins. Normal mode analysis (NMA) was the most preferable approach to the study dynamics of relatively big proteins and to predict large-scale motions of macromolecules of the proteins. With the introduction of coarse-grained techniques, NMA became more efficient in predicting both large and small amplitude of motions. For the current study, we employed WEBnm^@^, a new platform (http://apps.cbu.uib.no/webnma3, accessed on 8 September 2022) which calculates normal modes of proteins very efficiently utilizing a C-alpha force field for a few minutes and visualizing all results on the web portal without installing any additional plugins [48]. Single structure analysis and comparative analysis are available in this portal which makes the WEBnm^@^ the most advantageous option for the simulation studies. In this study, we utilized WEBnm^@^ for analyzing the stability and flexibility of the complexes formed between the mAb and the spike protein mutants of the Omicron variants. This advanced webtool visualizes vector fields and vibrations for each mode to represent motions and motion dynamics. WEBnm^@^ also calculates and compares deformation energies and fluctuations for each protein and plots them to reveal the flexibility and rigidity of the structures. Variations in deformation energy are due to the distance of C-alpha atoms from their neighboring ones. High deformation energies suggest flexible regions of the proteins such as loops, whereas low deformation energies indicate rigid regions i.e., helix. Herein, the comparative fluctuation plot defines the sum of atomic displacement of each low-frequency mode and is inversely related to their corresponding eigenvalues. The higher the eigenvalue directs the greater flexibility of the structure and a lower eigenvalue defines stable conformation of the complex. In the WEBnm^@^ result portal, the correlation matrix visualizes significant correlations among motions of C-alpha atoms and facilitates in analyzing the flexibility and rigidity of the complexes.

### 2.6. Designing and Characterization of Chimeric Antibody

After confirming the identity of the efficacious CDR patches present in the mAbs that have a strong binding affinity towards the spike protein mutants of Omicron variants, a series of chimeric antibodies were hypothesized to obtain better binding efficacy. In brief, the binding domain of the previously screened efficient mAbs was subjected to multiple sequence alignments and the peptide fragments showing distinct binding affinity against the mutant spikes of Omicron variants were combined *in silico*. The combined CDRs from different mAbs were taken in a single amino acid chain and the amino acid chain was subjected to antibody modelling using the therapeutic antibody profiler (TAP) web tool (http://opig.stats.ox.ac.uk/webapps/newsabdab/sabpred/tap, accessed on 15 september 2022) following Raybould et al. [49]. The modelled antibody was further characterized for its biochemical properties (solubility and hydrophobicity) and binding efficacy against the mutant Omicron strains through protein–protein interaction approaches described in the earlier section.

## 3. Results

### 3.1. Homology Modelling

Our study sought to determine the affinity of the 12 therapeutic mAbs viz. bamlanivimab, regdanvimab, tixagevimab, cilgavimab, etesevimab, casirivimab, imdevimab, sotrovimab, adintrevimab, beludavimab, lomtegovimab, and romlusevimab against the Omicron strains that harbour mutations in the spike protein. We first modelled the structure of each of the mAb using the respective amino acid sequences through homology modelling (Appendix A). Structures with optimum stereochemical quality, as demonstrated by Ramachandran plot, VERIFY3D, and Z-score (Appendix A), were selected for interaction studies.

### 3.2. Screening of the Potential mAbs

Molecular docking data clearly revealed the comparative efficacy of 12 mAbs against the native and mutant spike proteins of Omicron variants (Appendix A). The biophysical analyses of the protein–protein interactions and binding affinity between each mAb and viral spike protein revealed that all the mAbs exhibit strong binding affinity towards the native spike protein of SARS-CoV-2 while differential binding was documented against the mutant strains (Appendix A). The mAb-spike protein complex showing a Haddock 2.4 score of ≥100.00 and binding affinity of >15.0 kcal mol^−1^ was considered as a significantly interactive complex. Based on this selection criterion, five docked complexes comprising the five different mAbs, namely adintrevimab, beludavimab, regdanvimab, cilgavimab, and romlusevimab, were inferred to have intense neutralizing efficacy (Figure 1 and Table 1). Intriguingly, all of these five mAbs showed a strong binding affinity against the Omicron variants bearing G339D, S371L, G142D, N440K, and D614G mutations within the spike protein (Table 1). Our experimental data from the molecular docking study revealed that 74 spike protein–mAb complexes out of total of 315 protein complexes had binding affinity of more than −10.0 kcal mol^−1^, which primarily indicated that mAb therapy could be a strategy to treat Omicron.

### 3.3. Protein–Protein Interactions between mAb and Mutant Spike Protein(s) of Omicron

In order to study the molecular and biophysical mode of interactions, we selected the five stable spike protein mutant-mAb complexes such as adintrevimab_G339D, beludavimab_S371L, cilgavimab_G142D, regdanivimab_N440K, and romlusevimab_D614G. We first explored the involvement of noncovalent biomolecular interactions like hydrogen bonds, electrostatic bonds, and hydrophobic bonds, as these interactions are known to be the stabilizing backbone of the docked complexes and their presence determines the nature of the docked structures [30]. The involvement of different noncovalent bonds or forces resulting in different binding topologies and/or patterns are summarized in Table 2 and Figure 1. Among the five spike proteins of the mutant-mAb complexes, the cilgavimab_G142D structure possessed the maximum number of hydrogen bonds (total of 27 H-bonds), and this could be correlated with the strongest binding affinity (−18.6 kcal mol^−1^) of cilgavimab in comparison to the other mAbs studied here. Moreover, we found the involvement of a total of eight hydrophobic alkyl interactions that provided additional stability to the cilgavimab_G142D complex (Table 2).

Conformation of regdanivimab_N440K was found to be stabilized by a total of 23 hydrogen bonds and one hydrophobic alkyl interaction that primed to a strong binding affinity of −15.8 kcal mol^−1^ (Table 2 and Figure 1). Three other strongly bound docked structures viz. adintrevimab_G339D, beludavimab_S371L, and romlusevimab_D614G had a very similar binding affinity of −15.5 kcal mol^−1^, −15.0 kcal mol^−1^, and −15.3 kcal mol^−1^ respectively (Table 2 and Figure 1). Only 10 hydrogen bonds and a total of four hydrophobic bonds (two π–σ interaction, one π–π stacked and one π–alkyl interaction) were noted in the adintrevimab_G339D complex. The beludavimab_S371L complex possessed eight hydrogen bonds, one electrostatic bond, and five hydrophobic bonds (one π–π shaped, one alkyl interaction and three π–alkyl interaction). The last one, romlusevimab_D614G, also held a fewer number of hydrogen bonds (12 in number) along with two π–π stacked and three π–alkyl interactions. The binding topology and binding pattern of these three structures, i.e., adintrevimab_G339D, beludavimab_S371L, and romlusevimab_D614G, were totally dissimilar to that of the cilgavimab_G142D complex as well as to that of the regdanivimab_N440K complex and clarified the underlying reason for the differential binding affinity (Table 1 and Figure 1).

### 3.4. Molecular Dynamic Study to Verify the Stability

In addition to the biomolecular interactions, we have also evaluated the dynamics of the five selected spike protein mutant–mAb complexes in terms of their stability and compactness. A comparative dynamics study of adintrevimab_G339D, beludavimab_S371L, cilgavimab_G142D, regdanivimab_N440K, and romlusevimab_D614G complexes depicted that all the complexes were having lower average deformation energies that indicated more rigidity in the structures of the interacting proteins (Figure 2). In a comparative fluctuation plot, the presence of very few atomic displacements demonstrated greater stability and huge compactness within the spike protein mutant–mAb complexes (Figure 2). Moreover, stable conformational changes were also induced in both mAb and the mutant spike protein during the formation of the mAb–spike protein complex (Appendix A). Therefore, stability and stable binding conformation within the spike protein mutant–mAb complexes played the key role in mediating the efficacy of the mAbs. Whilst analysing each complex individually, we found the romlusevimab_D614G complex to have maximum lower average deformation energy in comparison to others, which suggested a strong association between romlusevimab and the D614G mutant spike protein (Appendix A). This prediction was further confirmed by the fact that the first nine modes of the romlusevimab_D614G complex were holding an eigenvalue of below 0.06, thus indicating that a lower deformation energy contributed to a huge stability of that complex (Appendix A). An abundance of correlated motions in C-alpha atoms of all five complexes (Appendix A) evidenced the presence of motion stiffness and rigidity in the complexes. Collectively, our *in silico* molecular dynamic study denoted adintrevimab, beludavimab, cilgavimab, regdanivimab, and romlusevimab for having a strong affinity to form biophysically stable conformations with G339D, S371L, G142D, N440K, and D614G respectively.

### 3.5. Chimeric mAbs and Their Potentiality as Immunotherapeutics

Our experimental data showed that adintrevimab, beludavimab, cilgavimab, regdanivimab, and romlusevimab could efficiently form stable complexes with the spike protein mutants of Omicron variants such as G339D, S371L, G142D, N440K, and D614G. However, there is a scope to increase the efficacy of the mAbs by combining the paratopes/CDRs of the active mAbs through generating chimeric antibodies. In this context, we observed regdanivimab, adintrevimab, and beludavimab as the effective mAbs for targeting the maximum number of spike protein mutants (Appendix A). Therefore, we combined the CDR regions of these mAbs and designed by an *in silico* a series of seven chimeric mAbs such as regdanivimab-framework-adintrevimab-CDRH3, adintrevimab-framework-regdanivimab-CDRH3, regdanivimab-framework-beludavimab-CDRH3, beludavimab-framework-regdanivimab-CDRH3, adintrevimab-framework-beludavimab-CDRH3, beludavimab-framework-adintrevimab-CDRH3, and sotrovimab-framework-regdanivimab-CDRH3 (Table 3).

Molecular docking-based protein–protein interactions and assessment of the binding affinity collectively evidenced the enhanced efficacy of the mAbs in the chimeric form and importantly all the seven chimeric mAbs were found to show a high binding affinity against the mutant spike protein of Omicron variants than that of their constituent mAbs (Table 4). While comparing the efficacy of all the chimeric mAbs against the Omicron variants, we found that chimeric mAb generated by the fusion of the framework of adintrevimab and CDRH3 of beludavimab (ARDYTRGAWFGESLIGGFDN) displayed a strong and stable binding with all the mutant forms of Omicron (Figure 3 and Table 4). Notably, this chimeric mAb showed very intense binding affinity (−16.0 kcal mol^−1^) with the seven spike protein mutants of the Omicron variants. Intriguingly, chimeric mAb was also found to be extremely stable in terms of stereo-chemical and biophysical attributes (Appendix A). We have also checked the efficacy of this hypothetical chimeric mAbs against the newly emerged Omicron strain BA.1, BA.2, BA.2.12.1 and BA.4/BA.5 (spike protein of BA.4 and BA.5 contain similar sequence) and the results collectively indicated that adintrevimab-framework-beludavimab-CDRH3 chimeric mAb could be a promising broad-spectrum anti-Omicron immunotherapeutic (Table 5).

## 4. Discussion

The conception of new therapeutic strategies, administration of several repurposed drugs, and different vaccines, along with the application of human mAbs, are indeed directing us to successful intervention of COVID-19 [50,51,52,53,54]. However, the available therapeutic strategies are continuously facing tremendous challenges due to the high mutability of SARS-CoV-2. In the recent past, we experienced the myriad threats posed by the mutant strains of SARS-CoV-2 like Alpha, Delta, and Delta-plus. Intriguingly, almost all the variants possess a mutation in the spike protein, the key protein in the pathogenesis of SARS-CoV-2. In this context, the recent emergence of Omicron variants and subvariants has been reported to have numerous mutations in their spike glycoprotein. Since the first report of its emergence on 24 November, 2021 in South Africa, more than 30-point mutations, including 15 substitutions, have been recorded within the RBD region of the spike glycoprotein of the Omicron variant B.1.1.529 [55]. All the mutations have been clinically correlated with the extremely high transmissibility of these variants [54,55]. Due to this attribute, new identities have been offered for Omicron as a variant of concern (VOC) and variant of interest (VOI) [56,57]. Moreover, Omicron variants are resistant or can escape neutralizing capacity of the antibodies elicited by either the available vaccine or from the convalescents [58,59,60]. A significant number of reports acquired from the different cohorts across the world also depicted that vaccination and/or infection-induced immunity is not effective to neutralize the Omicron variants [56]. Therefore, alternative therapeutics are needed to be screened on an urgent basis.

All the variants of SARS-CoV-2, along with Omicron, employ spike glycoprotein to invade the host cell [61,62]. In addition, the fatal pathological outcome, i.e., cytokine storm, is also induced through the binding of the spike protein to human TLR4 followed by NF-κB activation [19]. Considering this, targeting spike proteins could be the most beneficial way to develop immunotherapeutic agents like vaccines as well as mAbs against the newly emerged Omicron strains. In fact, several studies have demonstrated the efficacy of mAbs in neutralizing the Omicron variant having conserved RBD sequence [61]. The study conducted by Takashita et al. [63] revealed that REGN10987 (imdevimab) and COV2-2130 (cilgavimab) inhibit the viral load of Omicron variants BA.2.12.1, BA.4, and BA.5 while COV2-2196 (tixagevimab) was found to be effective against BA.2.12.1. On the other hand, bebtelovimab was considered as the most potent mAb against BA.4 and BA.5 variants and obtained emergency-use authorisation (EUA) in February 2022 from the Food and Drug Administration (FDA) [64]. However, due to less effectiveness of few mAbs against the newly emerged mutant variants of Omicron, the U.S. Department of Health and Human Services decided to stop the supply of them.

Keeping this in mind, we screened all the available mAbs for their binding capacity against the mutant spike proteins of Omicron and then customized the chimeric mAbs for conferring better efficacy against all of the reported Omicron variants having mutations in the RBD [65,66]. Our comparative protein–protein interaction study revealed a variable degree of binding efficacy of a total of 12 human mAbs against Omicron variants of SARS-CoV-2 (Appendix A). Omicron variants were found to have a greater affinity to human ACE2 due to the significant number of mutations in the spike glycoprotein, specifically in RBD [57]. Therefore, we tested all of the 12 mAbs against mutated spike glycoproteins utilizing *in silico* molecular docking and molecular simulation dynamics. Our previous study in this direction demonstrated regdanvimab, bamlanivimab, sotrovimab, etesevimab, and cilgavimab for having an excellent binding affinity towards Alpha and Delta variants of SARS-CoV-2 [30]. In the present study, we screened a total of 12 human mAbs including five mAbs that were previously studied by us [30] and seven additional mAbs viz. tixagevimab, casirivimab, imdevimab adintrevimab, beludavimab, lomtegovimab, and romlusevimab (Appendix A). The efficacy of each mAb was first tested against the native spike protein of SARS-CoV-2 and only nine (bamlanivimab, regdanvimab, cilgavimab, etesevimab, sotrovimab, adintrevimab, beludavimab, lomtegovimab, and romlusevimab) mAbs were found to exhibit a satisfactory level of binding (Haddock score < −85) (Appendix A). At the next level of screening, molecular docking was performed with the nine selected mAbs and 36 mutated spike proteins (among them, 35 mutants had a single mutation and a multiverse that had all the mutations).

The initial molecular docking-based protein–protein interaction studies figured out five mAbs namely adintrevimab, beludavimab, romlusevimab, cilgavimab, and regdanivimab that exhibit a high affinity (>−15.0 kcal mol^−1^) towards all the mutants of Omicron variants, especially against G339D, S371L, G142D, N440K, and D614G of Omicron variants (Table 1, Table 2 and Appendix A) and was further supported by binding affinity and molecular dynamics simulation studies. In fact, our experimental data on the dynamic behaviour of the mAb-spike protein complex predicted that the efficacious mAbs such adintrevimab, beludavimab, romlusevimab, cilgavimab, and regdanivimab possess a stereochemically and thermodynamically stable conformational topology while forming complexes with the mutant spike proteins (Figure 2 and Appendix A). Moreover, studies on the atomic displacement, fluctuation, and deformation energies collectively indicated that binding of therapeutic mAb to the mutant spike protein induces a conformational change that further stabilizes and strengthens the configuration and texture of the binding interface (Figure 2 and Appendix A). These can be considered as the underlying reasons behind the efficiency of each mAb to the mutants of Omicron variants (Figure 2 and Appendix A).

The above results prompted us to study the mode of biomolecular interactions between the mAbs and the mutant spike protein variants. Among the five active mAbs, cilgavimab and regdanivimab were also documented for having high binding affinity against Alpha and Delta strains, as shown in our earlier report [30]. Therefore, cilgavimab and regdanivimab could be considered for possible preclinical and clinical trials for their future use as therapeutics to combat the newly emerged variants of SARS-CoV-2, including the Omicron variants. Several studies have demonstrated the advantages of using the mAbs for treating COVID-19 patients owing to their target specific viral neutralizing efficiency, more specific binding, and limiting over secretion of inflammatory cytokines that have created a milestone in modern immunotherapeutic research [53,67]. Our *in silico* experiments demonstrate the affinity of human mAbs against Omicron variants and strongly advocate the candidature of cilgavimab and regdanivimab for combating the Omicron variants.

Hitherto, the major issue in dealing with the Omicron variants is that the mutant spike proteins could evade the action of antibodies and even the therapeutic mAbs. Interestingly, recent clinical trials conducted with the mAbs for treating SARS-CoV-2/Omicron infected patients depicted that combined therapy is more effective than monotherapy [68,69]. Another trial conducted by the Italian Medicines Agency further supported that combined therapy with Casirivimab and imdevimab has a preventing efficacy over Omicron infection [70]. Taking a cue from these studies, we hypothetically designed the chimeric mAbs by combining the CDR of the efficacious mAbs to improve their potential to neutralize the escaping mutants of the Omicron variants. A previous study on the application of experimentally developed chimeric antibodies that are composed of human IgG Fc region and mouse variable region has been found to confer a high degree of cross reactivity with Omicron BA.4/BA.5 [71]. Further, our earlier study showed an increased efficacy of chimeric mAb developed by fusing the CDRH3 of regdanvimab within the framework of sotrovimab against Alpha, Delta and Delta-plus strains of SARS-CoV-2 [30]. In this study, ligation of the CDR regions of the efficacious mAbs resulted in the generation of a total of seven stable chimeric mAbs to acquire better binding efficacy against the Omicron variants (Table 3). Our protein–protein interaction data clearly documented the enhanced binding efficacy the chimeric mAb generated by joining the CDRH3 of beludavimab within the framework of adintrevimab against all the Omicron variants BA.1, BA.2, BA.2.12.1, and BA.4/BA.5 (Table 4 and Table 5). This chimeric antibody could provide hope for treating the Omicron variants and the present study welcomes scientific validation in the near future. We have also included the *in silico* cloning for the recombinant production of the chimeric mAb in the bacterial system (Appendix A).

Advanced computational approaches are now in high demand to screen target antigens and proteins and to design therapeutic drugs, vaccines, and antibodies. In the previous work, we developed therapeutic chimeric mAb as a potential immunotherapeutic against the Alpha, Delta, and Delta-plus strains of SARS-CoV-2 by utilizing immunoinformatics and *in silico* structural biology [30]. Interestingly, modern bioinformatics is now accomplishing the homology modelling of murine antibodies and humanization of the same [72]. In this context, a study on the superhumanization of simian Fab 35PA83 (fragment of an antibody) exploiting the online program IMGT/V-QUEST and WAM tool have been found to reproduce the prediction regarding the neutralizing ability of the antibody when tested against lethal anthrax toxin [73]. In addition, a study by Khan et al. [74] also demonstrated the utility of computational techniques in designing the mAb variants in terms of upgraded binding potential. Moreover, computer-aided methodologies are now in use for analysing and improving the physiochemical stability of the mAbs [75]. Wolf Pérez et al. [76] successfully validated the *in silico* engineered mAbs through in vitro experimental studies. All of these findings strongly support the acceptability of the *in silico* assay in designing mAbs and diagnosing their efficacy. Up to today, the major drawback in conceptualizing the new therapeutic strategy for Omicron variants is the limited number of *in silico* studies that were performed in terms of searching the basic differences between the Omicron variants and other strains of SARS-CoV-2. In this direction, the present study adds a new dimension to the existing knowledge and developments pertaining to the combat strategies against Omicron. Our findings deciphered the effect of all the mutations on the structural attributes of the spike glycoproteins of SARS-CoV-2 and the identity of the possible therapeutic mAb to counteract them. Finally, the designed chimeric mAb, i.e., adintrevimab-framework-beludavimab-CDRH3 depicted in the study appears to be a promising immunotherapeutic to treat the Omicron variants.

## 5. Conclusions

As an effective treatment option, mAb therapy is rapidly emerging for a number of infectious, inflammatory, and autoimmune diseases in humans. Major pharmaceutical industries are now aiming towards the development of more advanced and target specific mAb therapies. Recent reports on the success of mAb-based therapies in treating COVID-19 patients prompted the scientific community to adopt this approach for intervening in the newly emerged strains of SARS-CoV-2. The recent emergence of Omicron variants bearing mutations within the spike glycoprotein, specifically in RBD, indeed put a huge challenge to most of the available treatment strategies. These strains are resistant to neutralization by the antibodies raised through vaccination and/or after the infection of SARS-CoV-2. In this context, our *in silico* study put forward the candidature of three human therapeutic mAbs, namely adintrevimab, beludavimab, and regdanivimab for treating Omicron variants. In addition, our *in silico* experiments also present the design of a new chimeric mAb by fusing the CDRH3 of beludavimab within the framework of adintrevimab for enhancing efficacy against all the Omicron variants, viz. BA.1, BA.2, BA.2.12.1, and BA.4/BA.5, through monotherapy.

However, our study has limitations. It was conducted through *in silico* approaches. Molecular docking is a good tool for predicting binding poses and as a starting point for molecular dynamic simulations. However, the docking scores do not express the quality of the different docked pose or how firmly the proteins are bonded to each other, specifically the binding potency of each interacting protein. Thus, molecular docking only predicts the different binding poses as per the lowest intermolecular energy and provides the docked conformations of the two proteins occurring in a complex. Such docked complexes consisting of the interacting proteins are useful for determining the biophysical basis of protein–protein interactions such as binding affinity, dynamicity of interaction, residual and atomistic fluctuation, molecular flexibility, and conformational changes. Therefore, exploiting only the molecular docking for screening and/or ranking the best-docked structure is not the appropriate way to examine the *in silico* protein–protein interaction events [77,78]. Nevertheless, our data provide a useful platform to researchers for experimental validation of the results.

## Figures and Tables

**Figure 1 antibodies-12-00017-f001:**
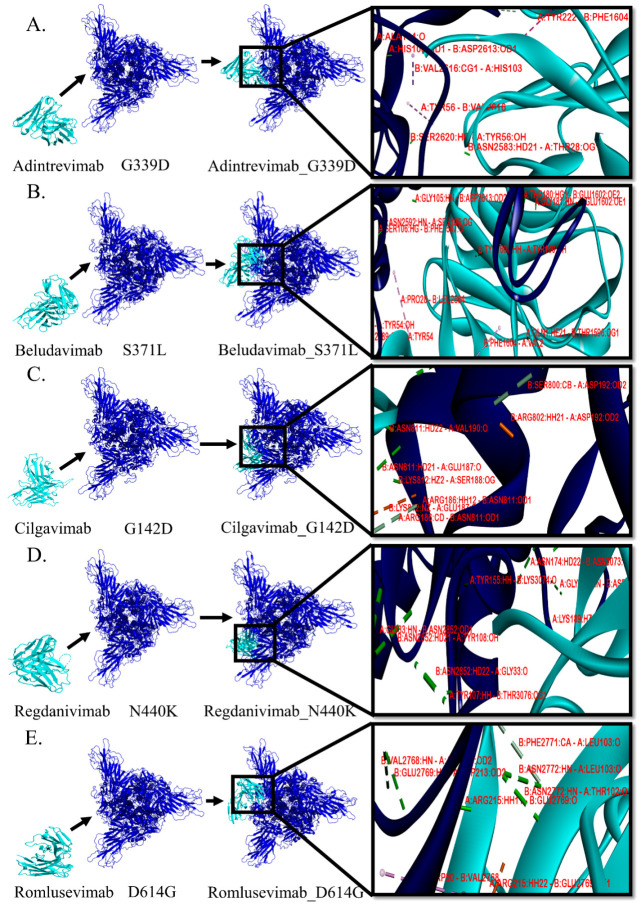
Comparative analyses on the binding pattern and binding topology of 5 mAbs–omicron spike protein complexes. (**A**) Interaction between adintrevimab and G339D, (**B**) beludavimab and S371L, (**C**) cilgavimab and G142D, (**D**) regdanivimab and N440K and (**E**) romlusevimab and D614G. (Blue colour is depicting the omicron variant and cyan for mAbs).

**Figure 2 antibodies-12-00017-f002:**
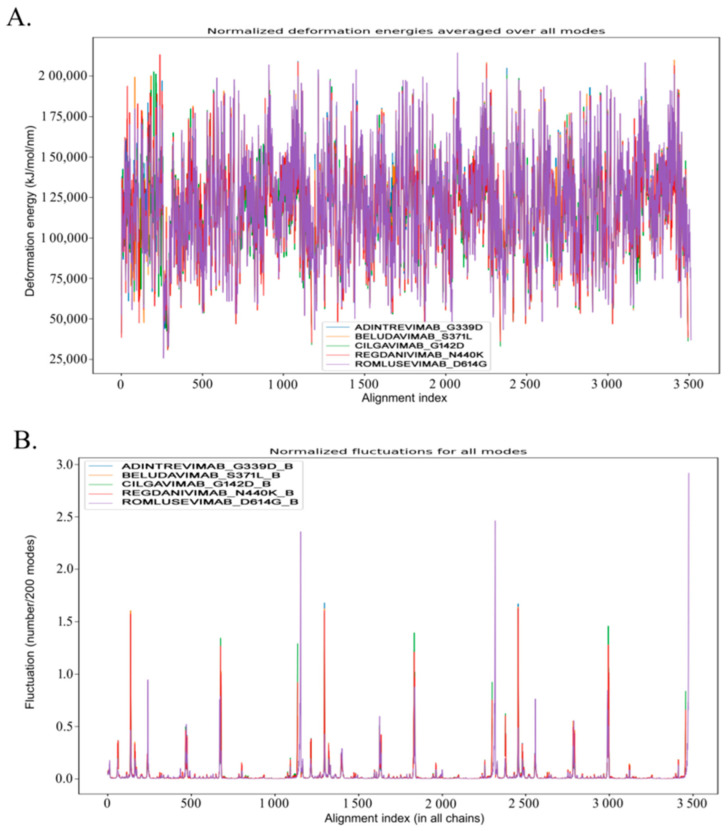
Analyses of the molecular dynamics of 5 mAb-Omicron spike protein complexes, adintrevimab_G339D, beludavimab_S371L, cilgavimab_G142D, regdanivimab_N440K, and romlusevimab_D614G, respectively. (**A**) Visualization of normalized deformation energies of 5 mAb–Omicron spike protein complexes. (**B**) Illustration of fluctuation details of the spike proteins of 5 mAb–Omicron spike protein complexes (Only B chain of the spike proteins was utilized to conduct the fluctuation study).

**Figure 3 antibodies-12-00017-f003:**
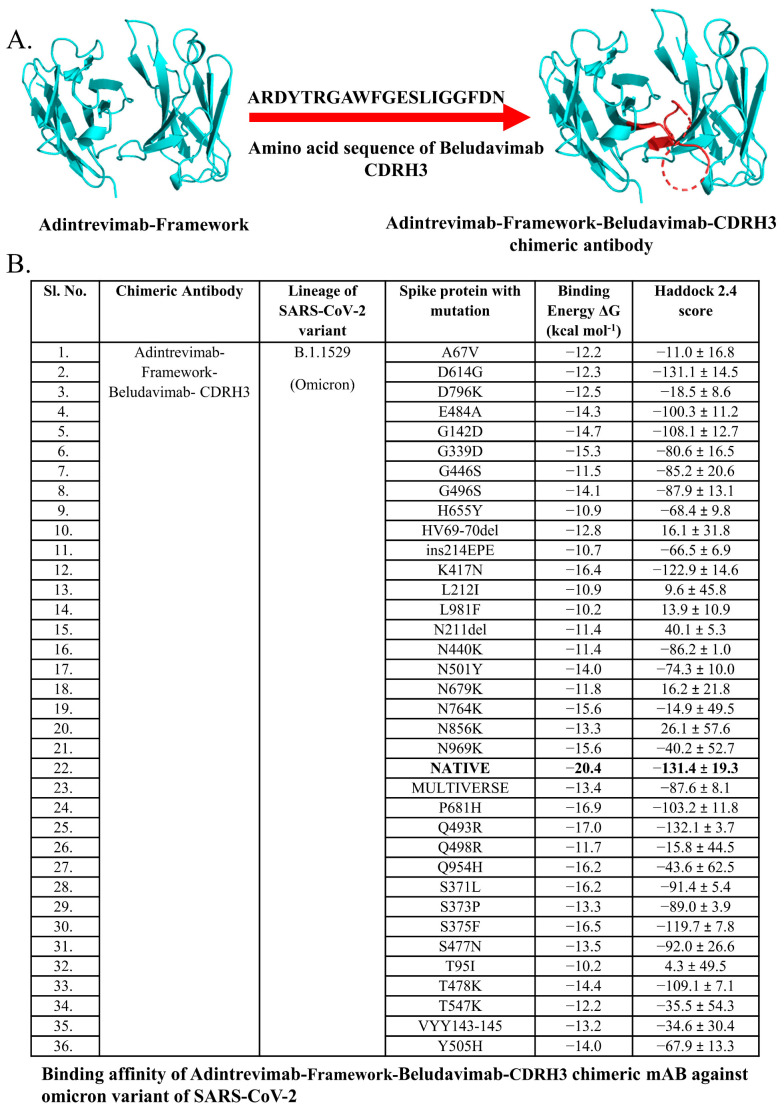
Predicted binding affinity of Adintrevimab-Framework-Beludavimab-CDRH3 chimeric mAb. (**A**) Structure of Adintrevimab-Framework-Beludavimab-CDRH3 chimeric mAb. (**B**) Predicted neutralizing efficacy of Adintrevimab-Framework-Beludavimab-CDRH3 chimeric mAb against omicron variant of SARS-CoV-2. Bold value is the best.

**Table 1 antibodies-12-00017-t001:** Selected mAbs and their level of binding efficacy against mutant spikes of B.1.1529 (Omicron) variant.

Sl. No.	Lineage of SARS-CoV-2 Strain	Spike Protein with Mutation	Interacting Monoclonal Antibody	Binding Affinity ΔG (kcal mol^−1^)	Haddock 2.4 Score
1.	B.1.1529 (Omicron)	G339D	Adintrevimab	−15.5	−113.7 ± 5.7
2.		S371L	Beludavimab	−15.0	−101.6 ± 4.9
3.		N440K	Regdanivimab	−15.8	−118.0 ± 17.7
4.		G142D	Cilgavimab	−18.6	−112.0 ± 14.4
5.		D614G	Romlusevimab	−15.3	−104.5 ± 3.7

**Table 2 antibodies-12-00017-t002:** Biomolecular interactions amongst the mutant spike proteins of Omicron variants and 5 selected efficacious mAbs.

mAb/Spike Protein Complex Structure	Hydrogen Bond			Electrostatic Bond			Hydrophobic Bond		
	mAb Residue	Spike Protein Residue	Distance (Å)	mAb Residue	Spike Protein Residue	Distance (Å)	mAb Residue	Spike Protein Residue	Distance (Å)
Adintrevimab/G339D	SER30	GLU2589	1.73	-	-	-	**π-σ interaction**		
	SER74	ASP2588	1.67	-	-	-	TYR214	THR1596	3.49
	HIS103	ASP2613	1.64	-	-	-	HIS103	VAL2616	3.93
	ALA104	ASP2638	1.72	-	-	-	**π-π stacked**		
	TYR154	CYS1598	2.13	-	-	-	TYR222	PHE1604	4.35
	THR28	ASN2583	1.96	-	-	-	**π-alkyl interaction**		
	TYR56	SER2620	1.73	-	-	-	TYR56	VAL2616	5.14
	ALA104	THR2634	3.18	-	-	-	-	-	-
	ALA104	ASN2637	1.97	-	-	-	-	-	-
	SER219	GLY1603	3.77	-	-	-	-	-	-
Beludavimab/S371L	GLN1	THR1596	1.97	TYR54	GLU2589	4.0984	**π-π shaped**		
	GLY105	ASP2613	1.70	-	-	-	TRP104	TYR2614	4.84
	SER106	PHE2587	1.76	-	-	-	**alkyl interaction**		
	THR180	GLU1602	1.95	-	-	-	PRO28	LEU2584	4.94
	GLY181	GLU1602	1.71	-	-	-	**π-alkyl interaction**		
	TYR100	TYR1607	2.37	-	-	-	TYR54	PRO2586	4.70
	SER106	ASN2592	2.30	-	-	-	TRP104	PRO2776	5.45
	TYR54	LYS2605	1.75	-	-	-	VAL2	PHE1604	5.26
	TPR104	PRO2776	3.02	-	-	-	-	-	-
Cilgavimab/G142D	LYS162	ASP817	1.53				**alkyl interaction**		
	ASP106	LYS291	2.42				PRO108	ILE1000	4.69
	ASP106	LYS291	1.62				PRO108	ILE2131	4.62
	ASP192	ARG802	1.84				PRO108	ILE3262	4.67
	GLU187	LYS812	4.52				ARG193	PRO799	4.82
	ASP106	LYS951	2.33				ALA201	ALA832	4.09
	VAL107	ARG1883	2.62				LEU156	ARG834	5.19
	GLN153	GLU268	2.31				PRO108	ALA2134	4.65
	SER154	GLU268	1.73				PRO108	ALA3265	5.34
	TYR157	THR294	2.48						
	SER158	SER33	1.83						
	ASN161	ASN947	2.83						
	LYS162	ASN940	2.56						
	ARG186	ASN811	2.37						
	ASP192	ASP807	2.66						
	SER159	THR294	2.56						
	GLU187	ASN811	2.90						
	VAL190	ASN811	1.80						
	SER188	LYS812	1.70						
	LEU109	LYS951	1.69						
	ASP106	ARG1883	1.96						
	ASP106	ARG1883	2.31						
	VAL107	THR1886	1.82						
	PRO108	ILE2131	2.41						
	ASP106	SER1876	2.85						
	ARG186	ASN811	2.99						
	ASP192	SESR800	3.39						
Regdanivimab/N440K	ASP56	LYS2527	1.59				**alkyl interaction**		
	LYS66	GLU2530	4.76				LYS66	ALA3094	4.95
	LYS66	ASP3092	5.10						
	GLY33	ASN2852	2.72						
	LYS59	ASN2529	1.80						
	TYR107	THR3076	1.84						
	TYR155	LYS3074	1.84						
	ASN174	ASN3073	2.83						
	LYS189	ASN3073	1.64						
	GLY191	ASP3069	2.39						
	ASP56	LYS2527	1.68						
	ASP57	THR2535	1.72						
	TYR108	ASN2852	3.06						
	GLY33	ASN2852	2.00						
	SER190	LYS3060	2.97						
	LYS189	ASN3073	2.89						
	ASN153	ALA3078	2.17						
	ASN153	LEU3077	1.91						
	ASN106	GLN3203	2.31						
	TYR105	THR3255	2.09						
	LYS66	GLU2530	3.57						
	LYS176	SER3189	3.18						
	ASN175	SER3188	3.24						
Romlusevimab/D614G	ARG215	GLU2769	2.44	TRP50	GLU2769	4.38929	**π-π stacked**		
	ASN52	GLU2769	2.14				TRP208	PHE2771	4.33
	SER210	THR2763	2.83				TRP208	PHE2771	5.27
	ARG215	GLU2769	1.78				**π-alkyl interaction**		
	TYR166	ALA359	1.80				TRP50	VAL2768	4.87
	SER75	LYS2729	1.65				TYR166	ALA359	4.64
	SER210	ASN2766	1.65				LEU103	TYR2774	5.43
	ASP213	VAL2768	2.87						
	ASP213	GLU2769	2.67						
	THR102	ASN2772	2.23						
	LEU103	ASN2772	2.62						
	LEU103	PHE2771	3.02						

All the bold lines are depicting the nature of the hydrophobic interaction.

**Table 3 antibodies-12-00017-t003:** List of chimeric mAbs * conceived by using the selected high affinity mAbs.

Sl. No.	Chimeric Antibody	mAb Framework	mAb CDR	CDR Sequence
1.	Regdanivimab-Framework-Adintrevimab-CDRH3	Regdanivimab	Adintrevimab	ARDFSGHTAWAGTGFEY
2.	Adintrevimab-Framework-Regdanivimab-CDRH3	Adintrevimab	Regdanivimab	ARIPGFLRYRNRYYYYGMDV
3.	Regdanivimab-Framework-Beludavimab-CDRH3	Regdanivimab	Beludavimab	ARDYTRGAWFGESLIGGFDN
4.	Beludavimab-Framework-Regdanivimab-CDRH3	Beludavimab	Regdanivimab	ARIPGFLRYRNRYYYYGMDV
5.	Adintrevimab- Framework-Beludavimab-CDRH3	Adintrevimab	Beludavimab	ARDYTRGAWFGESLIGGFDN
6.	Beludavimab-Framework-Adintrevimab-CDRH3	Beludavimab	Adintrevimab	ARDFSGHTAWAGTGFEY
7.	Sotrovimab-Framework-Regdanivimab-CDRH3	Sotrovimab	Regdanivimab	ARIPGFLRYRNRYYYYGMDV

* As variable degree of affinity was observed amongst the mAbs included in this study, the most active mAbs in terms of having high binding affinity were combined to prepare a chimeric antibody that can provide the best binding affinity to bind the spike proteins and neutralize the Omicron variants.

**Table 4 antibodies-12-00017-t004:** Comparative study on the binding affinity of different chimeric mAbs against omicron variants of SARS-CoV-2.

Mutant Spike Proteins of B.1.1529 (Omicron) SARS-CoV-2 Strain	Chimeric Antibody						
	Regdanivimab-Framework-Adintrevimab-CDRH3	Adintrevimab-Framework-Regdanivimab-CDRH3	Regdanivimab-Framework-Beludavimab-CDRH3	Beludavimab-Framework-Regdanivimab-CDRH3	Adintrevimab-Framework-Beludavimab-CDRH3	Beludavimab-Framework-Adintrevimab-CDRH3	Sotrovimab-Framework-Regdanivimab-CDRH3
	Binding Affinity ΔG (kcal mol^−1^)						
A67V	−6.7	−18.1	−10.0	−16.4	−12.2	−12.1	−16.3
D614G	−10.5	−10.1	−11.4	−11.5	−12.3	−14.1	−12.2
D796K	−6.6	−12.8	−10.9	−15.2	−12.5	−8.4	−14.2
E484A	−12.0	−14.6	−9.2	−10.1	−14.3	−14.8	−13.1
G142D	−11.9	−13.7	−11.5	−12.2	−14.7	−15.1	−10.5
G339D	−13.5	−9.8	−14.1	−12.2	−15.3	−12.5	−11.2
G446S	−10.8	−11.5	−14.5	−11.8	−11.5	−14.0	−11.9
G496S	−11.3	−12.4	−9.9	−10.8	−14.1	−14.0	−11.6
H655Y	−8.5	−15.3	−10.8	**−16.4**	−10.9	−10.1	**−16.8**
HV69-70del	−8.7	−15.3	−11.4	−10.4	−12.8	−11.7	−15.4
ins214EPE	−11.1	−15.6	−10.1	**−16.7**	−10.7	−12.9	−12.3
K417N	−11.1	−12.6	−12.2	−11.6	**−16.4**	−12.1	−11.5
L212I	−8.5	−13.2	−9.1	−13.6	−10.9	−10.9	**−17.8**
L981F	−11.7	−13.9	−10.6	−14.7	−10.2	−10.7	**−17.0**
N211del	−7.9	−15.6	−10.8	−14.6	−11.4	−13.6	−12.5
N440K	−12.2	−10.5	−11.7	−12.4	−11.4	−11.7	−13.7
N501Y	−12.6	−12.2	−9.2	−12.9	−14.0	−15.5	−12.3
N679K	−12.4	−14.6	−8.3	−13.0	−11.8	−13.8	−14.2
N764K	−9.9	−15.0	−11.2	−13.5	−15.6	−12.5	−10.5
N856K	−10.7	−13.9	−11.1	**−16.9**	−13.3	−12.1	−15.2
N969K	−10.6	−15.0	−10.5	**−16.3**	−15.6	−12.4	−13.6
NATIVE	−11.3	−10.6	−11.4	−9.9	**−20.4**	−10.3	−12.3
MULTIVERSE	−13.8	−11.7	−11.4	−11.8	−13.4	−12.4	−12.9
P681H	−13.7	−10.3	−14.8	−11.1	**−16.9**	−11.7	−11.4
Q493R	−13.8	−10.9	−13.0	−10.7	**−17.0**	−9.7	−11.3
Q498R	−10.9	−13.7	−12.3	−10.8	−11.7	−14.5	−12.6
Q954H	−7.2	**−16.3**	−10.6	−14.2	**−16.2**	−11.7	−14.1
S371L	−11.0	−11.5	−10.5	−12.0	**−16.2**	−12.1	−13.0
S373P	−11.9	−12.6	−10.4	−10.2	−13.3	−12.1	−10.9
S375F	−12.4	−10.8	−13.1	−11.8	**−16.5**	−10.3	−10.8
S477N	−12.7	−10.8	−11.9	−10.9	−13.5	−11.9	−10.9
T95I	−9.2	−11.4	−10.2	**−16.5**	−10.2	−7.6	−14.4
T478K	−12.7	−10.9	−12.6	−12.3	−14.4	−11.8	−12.6
T547K	−8.1	−14.1	−10.3	−15.5	−12.2	−14.0	−11.2
VYY143-145	−8.1	**−16.0**	−9.0	−13.6	−13.2	−7.9	**−16.3**
Y505H	−10.2	−14.0	−10.9	−10.3	−14.0	−9.5	−15.1

Bold values are depicting the strong binding affinities with ΔG < −16.0 kcal mol^−1.^

**Table 5 antibodies-12-00017-t005:** Affinity of chimeric mAbs against spike protein of omicron variants BA.1, BA.2, BA.2.12.1, and BA.4/5 of SARS-CoV-2.

Sl. No.	Chimeric Antibody	BA.1		BA.2		BA.2.12.1		BA.4/5	
		Haddock 2.4 Score	Binding Affinity ΔG (kcal mol^−1^)	Haddock 2.4 Score	Binding Affinity ΔG (kcal mol^−1^)	Haddock 2.4 Score	Binding Affinity ΔG (kcal mol^−1^)	Haddock 2.4 Score	Binding Affinity ΔG (kcal mol^−1^)
1.	Regdanivimab-Framework-Adintrevimab-CDRH3	−51.7 ± 5.4	-	−79.6 ± 14.1	-	−98.0 ± 8.8	-	−52.2 ± 2.6	-
2.	Adintrevimab-Framework-Regdanivimab-CDRH3	−98.3 ± 8.5	-	−115.1 ± 7.0	−14.0	−123.6 ± 5.9	−12.6	−92.5 ± 5.1	-
3.	Regdanivimab-Framework-Beludavimab-CDRH3	−52.1 ± 6.0	-	−51.4 ± 4.4	-	−72.0 ± 5.7	-	−62.9 ± 4.8	-
4.	Beludavimab-Framework-Regdanivimab-CDRH3	−96.0 ± 16.0	-	−145.4 ± 20.8	−15.1	−125.3 ± 21.1	−14.4	−102.0 ± 6.1	−13.1
5.	Adintrevimab-Framework-Beludavimab-CDRH3	**−111.6 ± 4.6**	**−15.0**	**−131.6 ± 7.1**	**−13.9**	**−135.4 ± 9.3**	**−17.4**	**−109.1 ± 12.0**	**−14.6**
6.	Beludavimab -Framework-Adintrevimab-CDRH3	−104.0 ± 7.1	−14.0	−118.2 ± 6.4	−12.6	−110.9 ± 2.8	−11.9	−77.7 ± 7.3	-
7.	Sotrovimab-Framework Regdanivimab-CDRH3	−105.1 ± 15.1	−15.8	−119.0 ± 4.3	−11.7	−109.2 ± 8.4	−12.0	−78.5 ± 12.7	-

Bold values are describing that the adintrevimab-framework-beludavimab-CDRH3 chimeric antibody is the best to form strong docked structure with all four omicron variants (BA.1, BA.2, BA.2.12.1, and BA.4/5).

## Data Availability

Data will be available on request to any of the corresponding authors.

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
