# Peer review of "Comparative Binding Ability of Human Monoclonal Antibodies against Omicron Variants of SARS-CoV-2: An In Silico Investigation"

_2073-4468, 2023, doi:10.3390/antib12010017_

Round 1
Reviewer 1 Report
In this work, the authors performed a pure computational modeling study to identify promising monoclonal antibodis targeting Omicron variants. Below are my comments that I hope the authors can address:
1) Since the whole work is completely based on computational predictions. It is important to convince readers including me that the methods/tools used for these predictions are reliable (as supported by previous work) especially considering the fact that there are many more rigorous methods available for these tasks. I hope the authors can clarify each method they used in this work for this purpose.
2) What is the reason of not showing atoms or ribbon-like conformation of the mAbs (Figure S1, 1, 3)? Only showing surface is not clear for atomic details.
3) The right column of Figure 1 is not clear at all. The authors need to figure out a better way to show these information.
4) Figure S3 and S4 are hard to read. Please prepare your figures with a high enough resolution.
Author Response
In this work, the authors performed a pure computational modelling study to identify promising monoclonal antibodies targeting Omicron variants. Below are my comments that I hope the authors can address:
Author’s response: Authors express their greetings to the learned reviewer for his/her valuable comments and propositions to improve the superiority of manuscript. We have tracked the recommendations provided by the reviewer and addressed all in the revised manuscript.
1) Since the whole work is completely based on computational predictions. It is important to convince readers including me that the methods/tools used for these predictions are reliable (as supported by previous work) especially considering the fact that there are many more rigorous methods available for these tasks. I hope the authors can clarify each method they used in this work for this purpose.
Author’s response: Authors express grateful thanks to the learned reviewer for his/her valuable comments and have incorporated all the necessary reliable details of methodology in the manuscript as per the suggestions of the reviewer.
We would like to inform you that our group is focused on in-silico structural biology and immune-informatics to present new therapeutics as well as intervention strategies to the scientific society for further development. In the present manuscript, we tried to present monoclonal antibodies as therapeutic options to treat the currently emerging and spreading form of SARS-CoV-2 i.e., Omicron variants. In this context, we used the very popular and efficacious protein-protein interactions strategy for elucidating the efficacy of mAbs against the mutated form of the viral spike proteins. We exploited molecular docking for determining the protein-protein interaction as it is a widely accepted and validated technique for examining the interaction between two proteins (Nadaradjane et al., 2018, https://doi.org/10.1007/978-1-4939-7759-8_28). The methodology is extremely easy and the process can be easily executed using both online and offline software. The methods for running molecular docking are available in every database and related literatures published by the different groups working on bioinformatics including us (Kangueane et al., 2018,
https://doi.org/10.1007/978-981-10-7347-2_14). Most importantly, we validated the postulations drawn from molecular docking using molecular dynamic simulation which now a days considered as the most scientifically validated tool for examining the computational predictions (Hollingsworth et al., 2018, https://doi.org/10.1016/j.neuron.2018.08.011). Our previous works in this direction revealed that computational prediction followed by MD study do simulate the actual experimental conditions. For example, our earlier work claiming SARS-CoV-2 spike proteins as the ligand of human TLR4 (DOI: 10.1002/jmv.26387) was successfully validated by most of the reputed workers work on COVID-19 immunology and this work has been cited with appreciation in more than 300 literatures till date. Therefore, the present study also promises to repeat the same.
2) What is the reason of not showing atoms or ribbon-like conformation of the mAbs (Figure S1, 1, 3)? Only showing surface is not clear for atomic details.
Author’s response: Authors express sincere thanks to the learned reviewer for his/her valuable comments and have incorporated the changes within the revised manuscript. There is no specific reason for using the surface like conformation of the mAbs in the manuscript. However, we have modified the Figures S1, 1 and 3 in ribbon-like conformation as per your kind suggestions.
3) The right column of Figure 1 is not clear at all. The authors need to figure out a better way to show this information.
Author’s response: Authors are thankful to the learned reviewer for his/her valuable suggestions and have incorporated the changes within the revised manuscript.
4) Figure S3 and S4 are hard to read. Please prepare your figures with a high enough resolution.
Author’s response: Authors expresses sincere thanks to the learned reviewer for his/her valuable suggestions and have incorporated the changes within the revised manuscript. As per the suggestions, we have upgraded the resolutions of the mentioned figures (S3 and S4).

Reviewer 2 Report
The authors have studied how different mAbs bind to different SARS-CoV-2 S protein variants. These calculations allowed them to determine the binding affinity and Haddock sore. They use these parameters to determine the therapeutic affinity of the mAbs against these proteins.
The introduction contains a series of mistakes, mostly on virological details. There are parts of the methodology that are poorly explained, and more importantly, they do not show how measuring the Gibbs free energy of mAb-S binding is related at all to "therapeutic affinity".
The data presented here does not help propose a measure of therapeutic activity and affinity, just binding. It is not clear that high binding affinity always results in therapeutic activity in all cases. Hence, either their claims are toned down, or they show how to relate these physicochemical parameters with a precise biological activity as therapeutic affinity. If they want to maintain their claim, they should correlate the proposed binding energies with experimental data of neutralization assays with the mAbs and variants.
I have uploaded a pdf file with all my comments.

Author Response
The authors have studied how different mAbs bind to different SARS-CoV-2 S protein variants. These calculations allowed them to determine the binding affinity and Haddock sore. They use these parameters to determine the therapeutic affinity of the mAbs against these proteins.
Author’s response: Authors express their greetings to the learned reviewer for his/her valuable comments and propositions to improve the superiority of manuscript. We have tracked the recommendations provided by the reviewer and addressed all in the revised manuscript.
The introduction contains a series of mistakes, mostly on virological details. There are parts of the methodology that are poorly explained, and more importantly, they do not show how measuring the Gibbs free energy of mAb-S binding is related at all to "therapeutic affinity".
The data presented here does not help propose a measure of therapeutic activity and affinity, just binding. It is not clear that high binding affinity always results in therapeutic activity in all cases. Hence, either their claims are toned down, or they show how to relate these physicochemical parameters with a precise biological activity as therapeutic affinity. If they want to maintain their claim, they should correlate the proposed binding energies with experimental data of neutralization assays with the mAbs and variants.
I have uploaded a pdf file with all my comments.
Author’s response: Authors are thankful to the to the learned reviewer for his/her valuable comments and propositions to improve the superiority of manuscript. The comments were really useful in improving the quality of the revised manuscript.
Regarding the comments on ‘virological details’ and ‘methodology’, we have revised the mentioned part in the revised manuscript accordingly and also elaborated the methodology section. The Gibb’s free energy is a thermodynamic parameter that measures the extent of protein-protein interaction between two unrelated proteins. The dynamics of complex formation by the two interactive proteins through physical interaction is a measure of the affinity of the two interacting proteins that tend to form a complex. This is a popular method for studying protein-protein interaction and used in several quality literatures that aim to present binding of one protein to other. (Lazim et al., 2020, https://doi.org/10.3390/ijms21176339).
We respect the comment of the learned reviewer that “The data presented here does not help propose a measure of therapeutic activity and affinity, just binding”. Actually, we do not wanted to claim the therapeutic activity as these monoclonal antibodies (mAbs) are already in use as therapeutics, rather we wanted to show that these mAbs can efficiently bind the mutated spike protein of the Omicron variants. In line with our in silico data, neutralizing ability of antibodies mounted by the ancestral strain of SARS-CoV-2 toward Omicron and other variants of concerns has been analysed recently by the one of the lead authors of the present manuscript (Chauvin et al., 2023, iScience, https://www.sciencedirect.com/science/article/pii/S2589004223002018 ). In fact, this work has demonstrated that Omicron virus were less sensitive to sero-neutralization by the antibodies mounted against ancestral strain of SARS-CoV-2
We also respect the other comment “It is not clear that high binding affinity always results in therapeutic activity in all cases.” Yes, high binding affinity always does not correlated with therapeutic activity rather only gives hints at the neutralizing ability of the mAbs.
Considering the reviewer suggestion and in agreement with the comment, we have toned down our claims (in the title, abstract and elsewhere in the article). The term ‘therapeutic activity’ has been removed in the revised text.
We have changed the focus more on binding than that of efficacy in the revised manuscript. The title of the manuscript has been modified as well. Lastly, all the comments of the learned reviewers were truly helpful in enriching and improving the quality of the revised manuscript and we express sincere thanks to the learned reviewer

Round 2
Reviewer 1 Report
In the revised manuscript, the authors address most of my comments. However, I do have one more comment that I hope the authors can incorporate in their manuscript before the acceptance of the manuscript.
The authors report docking scores along with binding affinities in multiple tables. It is important to highlight that a docking score is not equivalent to a binding affinity. The docking score can be used to rank docked poses but does not indicate how tight the compound can bind to the receptor. Many papers have proved that using docking scores to rank the compounds for their binding potency is not reliable. (see https://doi.org/10.1002/cmdc.202200425 and https://doi.org/10.1021/jm050362n) The authors should clarify this in the manuscript when talking about docking scores. The authors should also highlight this fact and cite these mentioned papers (or even more related work) to make sure readers do not mis-use docking scores in their own research. In general, docking is good for predicting binding poses and providing a starting point for molecular dynamics simulations. But docking is not an ideal tool to predict binding free energies and rank compounds for their binding potency.
Author Response
Reviewer 1:
In the revised manuscript, the authors address most of my comments. However, I do have one more comment that I hope the authors can incorporate in their manuscript before the acceptance of the manuscript.
The authors report docking scores along with binding affinities in multiple tables. It is important to highlight that a docking score is not equivalent to a binding affinity. The docking score can be used to rank docked poses but does not indicate how tight the compound can bind to the receptor. Many papers have proved that using docking scores to rank the compounds for their binding potency is not reliable. (see https://doi.org/10.1002/cmdc.202200425 and https://doi.org/10.1021/jm050362n) The authors should clarify this in the manuscript when talking about docking scores. The authors should also highlight this fact and cite these mentioned papers (or even more related work) to make sure readers do not mis-use docking scores in their own research. In general, docking is good for predicting binding poses and providing a starting point for molecular dynamics simulations. But docking is not an ideal tool to predict binding free energies and rank compounds for their binding potency.
Author’s response: Authors express sincere thanks to the learned reviewer for the valuable suggestion to improve the quality of the manuscript. We do agree with the reviewer that docking is good for predicting binding poses and as a starting point for molecular dynamics simulation. But is not an ideal tool to predict binding free energies and rank compounds for their binding potency. We have re-revised our manuscript accordingly.

Reviewer 2 Report
I want to thank the authors for taking into consideration all of my observations. The manuscript has been greatly improved; however, there are a few small details to be fixed that after being fixed the manuscript could be accepted:
1. Table 1. Please use two decimals for the H-bond distance.
2.- What are the units for the y-axis in figure 2?
3.- Figure 3. the binding energy column is missing to indicate the units
Author Response
Reviewer 2:
I want to thank the authors for taking into consideration all of my observations. The manuscript has been greatly improved; however, there are a few small details to be fixed that after being fixed the manuscript could be accepted:
Author’s response: Authors express their sincere gratitude to the learned reviewer for the valuable comments and propositions to improve the quality of the manuscript further. We have tracked the recommendations provided by the learned reviewer and addressed all of them in the 2nd revised manuscript.
- Table 1. Please use two decimals for the H-bond distance.
Author’s response: Thanks for the comment. As suggested, we have expressed the H-bond distances in two decimals in each case.
2.- What are the units for the y-axis in figure 2?
Author’s response: We have incorporated the unit of binding energy in figure 3 accordingly.
3.- Figure 3. the binding energy column is missing to indicate the units
Author’s response: We have incorporated the unit accordingly.
